# Molecular Mechanisms of Antiviral Agents against Dengue Virus

**DOI:** 10.3390/v15030705

**Published:** 2023-03-08

**Authors:** Michelle Felicia Lee, Yuan Seng Wu, Chit Laa Poh

**Affiliations:** Centre for Virus and Vaccine Research, School of Medical and Life Sciences, Sunway University, No. 5, Jalan Universiti, Bandar Sunway 47500, Selangor, Malaysia

**Keywords:** flavivirus, dengue virus, antiviral, small molecule, peptide, phytochemicals, mechanism of action

## Abstract

Dengue is a major global health threat causing 390 million dengue infections and 25,000 deaths annually. The lack of efficacy of the licensed Dengvaxia vaccine and the absence of a clinically approved antiviral against dengue virus (DENV) drive the urgent demand for the development of novel anti-DENV therapeutics. Various antiviral agents have been developed and investigated for their anti-DENV activities. This review discusses the mechanisms of action employed by various antiviral agents against DENV. The development of host-directed antivirals targeting host receptors and direct-acting antivirals targeting DENV structural and non-structural proteins are reviewed. In addition, the development of antivirals that target different stages during post-infection such as viral replication, viral maturation, and viral assembly are reviewed. Antiviral agents designed based on these molecular mechanisms of action could lead to the discovery and development of novel anti-DENV therapeutics for the treatment of dengue infections. Evaluations of combinations of antiviral drugs with different mechanisms of action could also lead to the development of synergistic drug combinations for the treatment of dengue at any stage of the infection.

## 1. Introduction

Dengue infections are caused by the dengue virus (DENV) and transmitted by *Aedes* mosquitoes [1]. Presently, dengue is endemic in 100 tropical and subtropical countries in Southeast Asia, Africa, the Americas, and certain European regions [2]. An estimated 390 million dengue infections and 25,000 deaths were reported annually, with 96 million cases exhibiting clinical manifestations [3,4]. DENV has four antigenically distinct serotypes, namely DENV-1 to 4. A primary infection with one serotype generates long-term immunity against that particular serotype and short-term immunity against the other three serotypes which lasts for about six months [5,6]. However, secondary infection with another DENV serotype might cause severe disease with complications leading to dengue haemorrhagic fever (DHF) and dengue shock syndrome (DSS) due to antibody-dependent enhancement (ADE) or original antigenic sin [7,8,9].

Vector control strategies such as fogging, rearing mosquito-eating fishes, clearing mosquito breeding habitats, utilizing larvicides to eliminate the larvae of *Aedes* mosquitoes, releasing genetically engineered *Aedes* mosquitoes into the community to reduce mosquito breeding, and infecting female *Aedes* mosquitoes with *Wolbachia* have been employed as efforts to reduce the population of *Aedes* mosquitoes, but these have not succeeded [10,11,12]. The only licensed dengue vaccine, Dengvaxia (CYD-TDV), is limited in usage due to a lack of DENV-1 and DENV-2 efficacies and its risk of causing severe dengue in dengue-naïve individuals [13,14,15].

To date, there are no clinically approved antivirals for the treatment of dengue infections, and treatment remains relying on supportive care such as fluid replacement and the use of analgesics [3]. A variety of antiviral drug candidates have not succeeded in reaching clinical trials due to poor physicochemical and pharmacokinetic properties [16]. Anti-DENV drug candidates such as chloroquine, prednisolone, lovastatin, and celgosivir have undergone clinical trials but failed to reduce viremia significantly to provide any significant beneficial effects [17,18,19,20]. Although various antiviral agents have been evaluated against DENV, most studies were performed mainly with in silico and in vitro analyses. The lack of in vivo evaluations in animal models could be a contributing factor to the absence of clinically approved DENV antivirals. This review highlighted the current development of antiviral agents against DENV in preclinical/clinical investigations, the different mechanisms of action employed by these antivirals against DENV, and the potential of combination therapy for the future development of DENV antivirals.

## 2. Structure and Genome Organization of DENV

DENV belongs to the genus *Flavivirus* within the *Flaviviridae* family [21]. The DENV virion is a spherical, enveloped particle which is 500 Å in diameter [22]. Its positive-sense RNA genome consists of 11 kilobases (kb) with a single open reading frame (ORF) characterized by the presence of two untranslated regions (UTRs) at both ends of the ORF. The 5′-UTR consists of 95–135 nucleotides with a type I cap-like mRNA whereas the 3′-UTR consists of 114–650 nucleotides and lacks a poly(A) tail, ending in a conserved stem-loop secondary structure (Figure 1) [23,24]. The main function of the 5′-UTR is to control gene expression by regulating translational efficiency. It affects the stability and localization of the mRNA whereas the 3′-UTR is made up of the conserved RNA structure vital for viral replication [25].

The RNA genome encodes for a polyprotein which undergoes proteolytic cleavage to generate three structural proteins, capsid (C), precursor membrane (prM) or membrane (M), envelope (E) proteins, and seven non-structural proteins (NS1, NS2A, NS2B, NS3, NS4A, NS4B, and NS5) [26,27]. The E and prM/M proteins are located on the DENV virion surface whereas the C protein which encapsulates the RNA genome is located below the lipid bilayer [28,29]. The structural proteins form the DENV viral particles whereas the non-structural proteins are essential components of the DENV life cycle for viral replication, assembly, maturation, and polyprotein cleavage [30,31,32,33].

## 3. DENV Life Cycle

A bite from a DENV-infected *Aedes* mosquito initiates the infection in humans. DENV can grow and replicate in various cells and organs including kidney, liver, spleen, and lymph nodes, but its major targets are dendritic cells, monocytes, and macrophages. The life cycle of DENV consists of several stages such as viral entry, replication, assembly, and release. First, DENV viral particles attach to the host cells via interactions between DENV surface proteins and their respective host receptors. The proposed host receptors that interact with DENV include dendritic cell-specific intracellular adhesion molecule-3-grabbing non-integrin (DC-SIGN), heparan sulfate, human C-type lectin-like molecule (CLEC5A), and mannose receptors. Studies have revealed that domain III of the DENV E protein is responsible for receptor recognition to allow DENV entry into host cells via receptor-mediated or clathrin-dependent endocytosis [34,35]. In addition, direct entry of DENV into host cells could also take place through the virus–host cell membrane fusion process [36,37,38].

After internalization, the low pH environment of the endosome causes the separation of E protein dimers on the surface of DENV virions. Subsequently, domain II of the separated E protein dimers hinges away from the virion surface, thereby allowing the fusion loop to insert itself into the endosomal membrane to bridge the viral and endosomal membranes [22,39,40,41]. This interaction promotes the formation of the E protein trimers, which is mediated by rearrangement of the domains [42]. During the trimerization process, domain III of the E protein rotates by 70° towards the endosomal membrane, forming interactions with domains I-II interfaces [40]. Next, the bending and hemifusion of both membranes is mediated by the backward folding of domain III to domain I and interactions between domain II and the stem region. This results in interactions between the fusion loop and the transmembrane domain to complete membrane fusion and the formation of pores. The pores allow the release of the DENV viral genome into the host cytoplasm [39,42,43].

Next, in the host cytoplasm, the released viral RNA is translated into a polyprotein. This is followed by the cleavage of the translated polyprotein by viral and host proteases to generate DENV structural and non-structural proteins. The non-structural proteins are essential during viral replication as they are responsible for further rounds of transcription to produce more viral genomes and proteins [44]. Viral assembly takes place at the endoplasmic reticulum (ER), whereby the ER membrane and viral glycoproteins envelope the C proteins and newly made viral RNA to form immature DENV particles. These immature DENV particles travel through the secretory pathway and the trans-Golgi network (TGN) for further processing of the prM protein by host furin proteases to form the mature M protein [45]. Following successful virus maturation, the mature DENV particles are then exocytosed from the host cells (Figure 2) [26].

## 4. Preclinical and Clinical Status of DENV Antivirals

Currently, there are no animal models which can reproduce the same symptoms or clinical manifestations of dengue infections as observed in humans. Various non-human primate models such as rhesus macaques, bonnet monkeys, olive baboons, and African green monkeys have been used for the development of vaccines or antibody-based therapies against dengue [46]. However, antiviral drugs are seldom evaluated in these non-human primate models due to issues such as high production costs, difficulty in breeding large numbers of animals, and the absence of severe dengue symptoms mirroring that in humans. On the other hand, murine models such as AG129 mice, BALB/c mice, and C57BL/6 mice have been widely used for the evaluation of in vivo antiviral drug efficacy against DENV. Wild-type immunocompetent murine models such as BALB/c mice and C57BL/6 mice pose various limitations including low systemic infections and the lack of severe dengue symptoms mirroring that in humans. AG129 mice which are deficient in IFN-α/β and γ receptors are more commonly used for antiviral drug testing against DENV, although they also exhibit a lack of severe dengue symptoms mirroring that observed in humans [47,48].

The use of mouse-adapted DENV and non-mouse adapted DENV strains which cause high virulence in AG129 mice had resulted in disease manifestations similar to dengue infections in humans such as increased levels of pro-inflammatory cytokines, high viremia, vascular leakage, and thrombocytopenia [49]. The in vivo efficacy of antiviral drugs against DENV could be determined by evaluating the reduction in viremia or protection from mortality in AG129 mice. Although many antiviral drugs were reported to possess good anti-DENV activities in vitro, not all of them had been evaluated for their in vivo efficacy in a suitable animal model [48]. Examples of compounds that had been evaluated for their in vivo efficacy against DENV include geraniin, NITD-618, PG545, and HS-1 [50,51,52,53]. The lack of in vivo evaluations may also contribute to the lack of DENV antivirals undergoing clinical trials. To date, there are very limited DENV antivirals that are undergoing clinical trials. The list of DENV antivirals that are currently undergoing clinical trials are presented in Table 1.

## 5. Modes of Action of Antivirals

To date, several approaches have been explored for the development of antiviral agents against DENV by targeting host attachment factors or cell receptors, DENV structural and non-structural proteins, as well as stages during post-infection such as viral replication, viral maturation, and viral assembly. Antivirals designed to target host attachment factors or cell receptors might prevent the binding and attachment of DENV viral proteins with the host cell, thereby inhibiting entry of DENV into host cells. On the other hand, antivirals that target DENV structural proteins might interfere with the binding of DENV to host cells, thereby preventing viral entry into host cells. Antivirals that target DENV non-structural proteins could interfere with viral replication as these proteins are vital components of the DENV replication machinery [28].

### 5.1. Host-Directed Antivirals

An approach which has been utilized to inhibit DENV infections is blocking host attachment factors or cellular receptors to prevent DENV attachment and entry into host cells. Several molecules identified as possible host cellular receptors or attachment factors for DENV include DC-SIGN (also known as CD209), heparan sulfate, CD14, HSP90/HSP70, claudin-1, glucose regulated protein 78 (GRP78), mannose, laminin, integrins, and the TIM and TAM proteins (Figure 3) [54,55,56,57,58,59,60,61,62,63]. However, the major drawbacks of host-directed antivirals include cellular side effects and cytotoxicity as the targeted host receptors or attachment factors might be essential for host cell survival [64]. The identified host-directed antivirals of DENV have been summarized in Table 2.

#### 5.1.1. DC-SIGN

DC-SIGN is a C-type lectin, type II transmembrane protein containing an extracellular domain which interacts with mannose-containing carbohydrates. DC-SIGN takes part in various processes including antigen recognition and presentation, priming of T-cells, and migration of dendritic cells [65]. It was identified to be an important cellular factor for effective DENV infection of immature dendritic cells [59,66]. The mannose N-glycan groups located on the DENV E protein mediate the interactions between DENV viral particles and DC-SIGN [67]. The DENV E protein glycosylated residue Asn67 functions as the DC-SIGN binding site [68]. It was found that the interactions between DC-SIGN and DENV E protein occurred due to hydrogen bonds between mannose molecules of Asn272 of DC-SIGN and Asn67 of DENV E protein as well as through salt bridges formed between certain amino acid residues of DC-SIGN and DENV E protein [69]. Therefore, blockage of the binding of DENV to DC-SIGN might prevent DENV infection.

Currently, limited antivirals were found to prevent DENV infections by blocking the binding of DENV to DC-SIGN (Table 2). Chen et al. (2017) showed that bovine lactoferrin was able to confer protection to DC-SIGN-expressing THP-1 cells against DENV-2. The infection rates of DENV-2 were found to be significantly higher in DC-SIGN-expressing THP-1 cells when compared to DC-SIGN-deficient THP-1 cells, thus indicating the vital role of DC-SIGN in DENV-2 infections. However, upon addition of bovine lactoferrin at 200 μg/mL, the infection rates of DENV-2 in DC-SIGN-expressing THP-1 cells were markedly reduced. On the other hand, the presence or absence of bovine lactoferrin does not affect the infection rates of DENV-2 in DC-SIGN-deficient THP-1 cells. This indicated that bovine lactoferrin exerted its inhibitory effects against DENV-2 by blocking the interactions between DENV-2 viral particles and the DC-SIGN receptor [70]. Alen et al. (2011) demonstrated the antiviral effects of carbohydrate-binding agents (CBAs)—*Hippeastrum hybrid* (HHA), *Urtica dioica* (UDA), and *Galanthus nivalis* (GNA) against all four DENV serotypes in Raji/DC-SIGN^+^ cells and monocyte-derived dendritic cells (MDDC). Interestingly, these CBAs demonstrated 100-fold higher anti-DENV activity in MDDC than in Raji/DC-SIGN^+^ cells. It was shown that the binding of HHA to the DENV-2 E protein blocked the interactions between DENV-2 and DC-SIGN, subsequently interfering with DENV-2 attachment to the host cells. This indicated that these CBAs acted during an early stage of DENV infection by preventing viral attachment to DC-SIGN receptors on host cells [71].

#### 5.1.2. Heparan Sulfate

Heparan sulfate (CD44) is classified under the glycosaminoglycans (GAGs) family, and it is made up of an O-sulfated glucosamine derivative and disaccharide chains with L-iduronic or uronic acids [72]. It was discovered that the heparan sulfate receptor interacts with DENV through the GAG-binding sites on the DENV E protein, whereby the positive-charged residues on DENV E protein bind to the negative-charged residues on heparan sulfate [55,73,74]. A study has also identified that conserved residues Lys291 and Lys295 are vital for the interactions between the DENV E protein and heparan sulfate [75]. Therefore, blockage of the binding of DENV to heparan sulfate might prevent DENV infection.

Various studies have identified antivirals that prevent DENV infections by blocking DENV-heparan sulfate binding (Table 2). Chen et al. (2017) has shown that bovine lactoferrin was able to confer protection to heparan sulfate-expressing Chinese hamster ovary (CHO)-K1 cells against DENV-2. The infection rates of DENV-2 were found to be significantly higher in heparan sulfate-expressing CHO-K1 cells when compared to heparan sulfate-deficient CHO-pgsA745 cells, thus indicating the vital role of heparan sulfate in DENV-2 infections. However, upon the addition of bovine lactoferrin at 0.1–200 μg/mL, the infection rates of DENV-2 in heparan sulfate-expressing CHO-K1 cells were significantly reduced in a dose-dependent manner. The presence or absence of bovine lactoferrin did not affect the DENV-2 infection rate of heparan sulfate-deficient CHO-pgsA745 cells. This indicated that bovine lactoferrin exerted its inhibitory effects against DENV-2 by blocking the interactions between DENV-2 viral particles and the heparan sulfate receptor [70]. Next, Recalde-Reyes et al. (2022) investigated the anti-DENV activities of a linear antiviral peptide, PD1 CD44, which was derived from the heparan sulfate receptor. The PD1 CD44 peptide was able to effectively inhibit DENV-1 to 4 infections by binding to domain III of the DENV E protein. This interaction presumably blocked the interaction between the DENV E protein and the CD44 receptor, thus resulting in inhibition of DENV-1 to 4 infections [76]. A heparan sulfate mimetic, PG545, potently inhibited DENV-2 with an IC_50_ value of 25 nM. The IC_50_ value referred to the half-maximal inhibitory concentration required to inhibit the virus. It also blocked the induction of cellular activation and NS1-mediated disruption of endothelial monolayer integrity. Additionally, PG545-treated AG129 mice demonstrated improved survival rates with lower viremia, serum TNF-α, circulating NS1, and systemic vascular leakage [52].

#### 5.1.3. Other Receptors

Integrins are made up of α and β heterodimeric subunits and they are essential for cell migration, adhesion, and recognition of extracellular matrix proteins. The most abundant surface receptors of vascular endothelial cells (VECs) and platelets are β1 and β3 integrins, and they are important for maintaining the permeability and integrity of capillaries [77]. Studies have found that DENV-2 infection enhanced β3 integrin expression in VECs. Pre-incubation of DENV-2 with soluble β3 integrin and down-regulation of β3 integrin expression from RNA interference was found to strongly inhibit DENV-2 entry into VECs, which indicated the possible role of β3 integrin receptor for DENV-2 entry into VECs [62,63]. However, very limited antivirals were found to prevent DENV infections by blocking the binding of DENV to β3 integrin (Table 2). Cui et al. (2018) evaluated several peptides which were designed based on domain III of the DENV E protein and it was found that P4 and P7 could effectively block DENV-2 entry into human umbilical vein endothelial cells (HUVECs) via interactions with β3 integrin. The authors proposed that peptides P4 and P7 blocked DENV-2 entry into HUVECs by occupying the binding site of DENV on the β3 integrin receptor [78].

Apart from that, the TIM and TAM receptors were also identified to play a role in DENV cellular attachment. These receptors were reported to be involved in the phagocytosis and elimination of apoptotic cells via recognition of the phosphatidylserine (PtdSer) apoptotic marker. During a DENV infection, the viral membrane exposed PtdSer, and this enabled virus entry into host cells by direct binding to the TIM receptor or indirect binding to the TAM receptor via the PtdSer binder molecule, Gas6. The activation of the TAM receptor played an essential role in DENV pathogenesis as it prevented the expression of genes involved in the interferon pathway, which is important for cellular antiviral response [56,79,80]. However, to date, there are no known antivirals that could inhibit DENV infection by blocking interactions between DENV and the TIM and TAM receptors. Similarly, no antivirals were identified to block DENV infection by targeting other host receptors such as mannose, laminin, HSP90/HSP70, GRP78, CD14, and claudin-1 [54,56,57,58,60]. Inhibitors targeting host receptors are known to be less prone to the development of resistance as opposed to those that directly target viruses, and this might serve as a good strategy to develop DENV inhibitors.

Although most studies demonstrated that the entry of DENV into host cells occurred via receptor-mediated or clathrin-dependent endocytosis; entry of DENV into host cells via clathrin-independent endocytosis has also been shown [81,82,83]. Direct entry of DENV into host cells via virus–host cell membrane fusion resulting in direct DENV penetration into the host cytoplasm without endocytosis has also been observed [84]. In addition, various studies have also found that DENV infects various cell types including cells from humans, hamsters, mosquitoes, and monkeys via various receptors [66,85,86,87,88,89]. The pathway for the entry of DENV into host cells depends on the type of cell and the DENV strain. Due to the wide range of DENV entry routes into host cells, DENV host cellular receptors, and its broad tissue tropism, DENV structural or non-structural proteins are easier targets for the development of DENV inhibitors [28].

**Table 2 viruses-15-00705-t002:** Host-directed antivirals of DENV.

Drug	Target(s)	Mechanism(s) of Action	Inhibitory Activities (IC_50_/EC_50_ Values)	Reference
Bovine lactoferrin	DC-SIGN, heparan sulfate	Inhibition of the binding of DENV to DC-SIGN and heparan sulfate	IC_50_: D1-165.8 ± 35.1 μM D2-40.7 ± 8.6 μM D3-166.7 ± 30.6 μM D4-164.5 ± 41.0 μM	[70]
*Hippeastrum hybrid* (HHA)	DC-SIGN	Inhibition of the binding of DENV to DC-SIGN	EC_50_ against DENV-2: HHA-4.6 nM UDA-3.8 nM GNA-480 nM	[71]
*Urtica dioica* (UDA)				
*Galanthus nivalis* (GNA)				
PD1 CD44	Heparan sulfate	Inhibition of the binding of DENV to heparan sulfate	IC_50_ against DENV-2: 13.8 μM	[76]
PG545			IC_50_ against DENV-2: 25 nM	[52]
Fucoidan			IC_50_: D1->1000 μg/mL D2-4.7 μg/mL D3-500 μg/mL D4-365 μg/mL	[90]
PI-88			EC_50_ against DENV-2: 200 μg/mL	[91]
dl-galactan hybrid C2S-3			IC_50_: D2-1 μg/mL D3-13.9–14.2 μg/mL D4-29.3->50 μg/mL	[92]
iota-carrageenan G3d				
CF-238			IC_50_: D1-24 μM D2-46 μM D3-14 μM D4-47 μM	[93]
Sulfated galactomannan	Heparan sulfate	Inhibition of the binding of DENV to heparan sulfate	IC_50_ against DENV-2: 0.12-20 μg/mL	[94]
Sulfated galactan				
Curdlan sulfate			EC_50_: D1->0.262 μg/mL D2-7 μg/mL D3-0.01 μg/mL D4-0.069 μg/mL	[95]
Chondroitin sulfate E			EC_50_: D1-0.53 ± 0.10 μg/mL D2- 3.80 ± 0.68 μg/mL D3-1.38 ± 0.33 μg/mL D4-0.30 ± 0.06 μg/mL	[96]
P4	β3 integrin	Inhibition of the binding of DENV to β3 integrin	IC_50_ against DENV-2: 19.08 ± 2.52 μM	[78]
P7			IC_50_ against DENV-2: 12.86 ± 5.96 μM	

The IC_50_ value referred to the half-maximal inhibitory concentration required to inhibit the virus. The EC_50_ value referred to the concentration of the compound that gave the half-maximal response.

### 5.2. Direct-Acting Antivirals

Direct-acting antiviral agents referred to antivirals that interact directly with viral proteins to exert their antiviral functions [97]. They offered a more promising approach as compared to host-directing antiviral agents as they target a specific viral protein, and this might offer a wide spectrum of antiviral activity and low toxicity. However, a major drawback of direct-acting antiviral agents is the risk of developing resistance [98]. The most broadly studied DENV structural protein is the E protein whereas the most studied DENV non-structural proteins are the NS3 and NS5 proteins [99]. The identified direct-acting antivirals of DENV have been summarized in Table 3 and Table 4. The chemical structures of selected direct-acting antivirals of DENV have been summarized in Table 5.

#### 5.2.1. Targeting DENV E Protein

The DENV E protein is a 53 kDa class II viral membrane fusion protein [22]. It forms dimers in mature DENV virions and trimers in immature DENV virions or fusion intermediates. In mature DENV virions, the antiparallel E protein dimers lie flat on the virion surface. On the other hand, in immature DENV virions and fusion intermediates, the E protein trimers point away from the viral membrane, and this gives the virion a spiky appearance [100].

The N-terminal region of the DENV E protein consists of domains I, II, and III whereas the stem region and transmembrane (TM) domain are located in its C-terminal region. Domain I or EDI is the central domain which stabilizes the overall orientation and engages in conformational changes of the E protein [101,102]. Domain II or EDII contains the highly conserved fusion loop (amino acids 98–112) which is responsible for the homodimerization process of the E protein [102]. Next, domain III or EDIII is known as the receptor-binding domain which is essential for receptor-binding [22,103]. It contains important antigenic epitopes for direct interactions with potent neutralizing epitopes and receptor-binding sites that accommodate viral entry into host cells [104,105]. The stem region is linked to the N-terminal domains via domain III, and it is made up of α-helices, E-H1 and E-H2, which are linked together by a highly conserved sequence [106]. It rearranges the positions of its two α-helices during the membrane fusion process. The TM domain is made up of antiparallel helices, TM1 and TM2, which are important for the completion of viral–host cell membrane fusion and stabilization of the trimeric E protein [107].

Several studies have identified various antivirals which inhibit DENV infections by targeting the E protein (Table 3). The anti-DENV activities of dipeptide, EF, that could occupy the hydrophobic site of the DENV E protein were investigated. It significantly reduced DENV-2 foci formation by 90% with an IC_50_ value of 96.50 μM. It also lowered the production of DENV-1, DENV-3, and DENV-4 E proteins by 20–40%, though a more significant drop in the production of DENV-2 E protein (70%) was observed [108]. Hrobowski et al. (2005) evaluated the anti-DENV activities of peptides which were derived from the DENV E protein sequences and found that peptide DN59 was the most potent peptide with a significant inhibition of DENV-2 (>99%) plaque formation when administered at concentrations <25 μM. At 20 μM, it also demonstrated a 100.0 ± 0.5% maximum inhibitory activity against DENV-2 [109]. Schmidt et al. (2012) showed that compound 1662G07 and its analogues were able to inhibit DENV infectivity by binding to the DENV E protein and subsequently prevented viral attachment. These compounds were found to bind to the β-OG pocket in the E dimer, and this blocked dimer-to-trimer transition, resulting in the inhibition of low-pH triggered DENV-liposomes fusion [110]. A study by Poh et al. (2009) reported a DENV fusion inhibitor, NITD448, that could potentially bind to the β-OG pocket in the DENV E protein. NITD448 effectively inhibited DENV-2 with an EC_50_ value of 9.8 μM, and its antiviral effects were only observed during the early viral entry stage. The EC_50_ value referred to the concentration of the compound that gave the half-maximal response. Docking studies suggested that the carboxylate group on the chromenone ring of NITD488 interacted with Gln52 and Lys128 of the E protein, with the trifluoro-phenyl motif of the molecule being buried deeply in the β-OG pocket [111].

#### 5.2.2. Targeting DENV prM/M and C Proteins

Although most studies are concentrated on the DENV E protein due to its importance in viral–host cell membrane fusion and receptor-binding, the prM/M and C proteins are also suitable targets for the development of anti-DENV therapeutics (Table 3). The DENV prM is a 21 kDa protein which is the precursor of the 8 kDa M protein. Host furin proteases cleave the prM protein to separate it into the pr peptide (amino acids 1–91), ectodomain (amino acids 92–130), and M protein (amino acids 131–166) [112,113,114]. The prM protein functions to prevent the conformational changes of the E protein during virus maturation in the low pH environment of the TGN [28]. In a study by Panya et al. (2015), MLH40, a peptide inhibitor that could mimic the conserved ectodomain of DENV M protein was investigated. Peptide MLH40 exhibited potent inhibitions of DENV-1 to 4 with IC_50_ values of 24–31 μM. MLH40 was able to bind to the inner site of the DENV E homodimer, which was the interacting site between DENV M and E proteins [115]. Another study has revealed that the pr peptide could bind to the DENV E protein at low pH, resulting in inhibition of viral–host membrane fusion and subsequent infection. Substitution of the highly conserved H244 residue at the pr-E interface with alanine resulted in the blockage of pr-E interaction and a reduced release of DENV virus-like particles (VLPs), which indicated that the pr peptide interacted with the DENV E protein via the H244 residue. The blockage of the pr-E interaction could have caused a distortion in the conformation of the DENV E protein, which could potentially lead to an interference with the pH-dependent fusion reaction in the endosome [116].

The 11 kDa DENV C protein consists of four α-helical regions with antiparallel arrangements of its homodimers. It also contains a high net charge with an asymmetrical arrangement of basic residues lying along its surface which functions to orchestrate the binding of RNA. On the other hand, it also has a hydrophobic region which interacts with lipids [117,118]. This structure makes the C protein an important component in virus assembly as it facilitates encapsulation of the RNA genome for nucleocapsid formation [113]. In addition, the hydrophobic region of the C protein contains a signal sequence which anchors the C protein and partitions the prM protein to the ER membrane [114]. During virus assembly, the NS2B-NS3 protease cleaves off the C-terminal to form a mature protein [117]. In a study by Faustino et al. (2015), pep14-23 derived from a conserved region of the DENV C protein inhibited interactions between the DENV C protein and host intracellular lipid droplets which was vital for DENV replication [119]. Apart from that, a small molecule, ST-148, interacted with the DENV C protein to inhibit virus assembly and release. ST-148 potently inhibited DENV-2 with EC_50_ value of 0.016 μM and EC_90_ value of 0.125 μM. It also potently inhibited DENV-1, DENV-3, and DENV-4 with EC_50_ values ranging from 0.512–2.832 μM. ST-148 showed high inhibition when it was added up to 12 h post-infection which suggested that it exerted its antiviral effects during the post-entry stage. The authors proposed that ST-148 exerted its antiviral effects by enhancing interactions with the DENV C protein to induce structural rigidity and steric hindrance during nucleocapsid formation, thereby inhibiting virus assembly and release [120,121]. Two DENV small molecule inhibitors, VGTI-A3 and VGTI-A3-03, were also found to block DENV infections by binding to the DENV C protein. VGTI-A3 potently inhibited the production of infectious DENV-2 progenies with an IC_90_ value of 112 nM, but its low solubility made it unsuitable to be developed as a DENV antiviral. Its analog, VGTI-A3-03, demonstrated higher antiviral effects against DENV-2 with an IC_90_ value of 25 nM and improved solubility, making it a more suitable DENV antiviral candidate. VGTI-A3 demonstrated specific antiviral activity against DENV-2 only whereas VGTI-A3-03 exhibited inhibitions of DENV-1, DENV-4, and West Nile virus (WNV). VGTI-A3-03 was found to interact with a binding pocket at the dimerization interface of the DENV C protein. This binding occurred during DENV virion formation, and this led to the incorporation of VGTI-A3-03 in the viral particle, resulting in formation of non-infectious DENV particles [122].

**Table 3 viruses-15-00705-t003:** Direct-acting antivirals of DENV targeting DENV structural proteins.

Drug	Target(s)	Mechanism(s) of Action	Inhibitory Activities (IC_50_/IC_90_/EC_50_ Values)	Reference
1662G07 and analogs	E protein	Fusion inhibitor	IC_90_: D2-0.89–2.0 μM D4-1.3–2.3 μM	[110]
DN59			IC_50_ against DENV-2: ~10 μM	[109]
NITD448			IC_50_ against DENV-2: 6.8 μM EC_50_ against DENV-2: 9.8 μM	[111]
DV2^419–447^			IC_90_: D1-0.1 μM D2-0.3 μM D3-2 μM D4-0.7 μM	[123,124]
DN57opt			IC_50_ against DENV-2: 8 ± 1 μM	[125]
1OAN1			IC_50_ against DENV-2: 7 ± 4 μM	
Rolitetracycline			IC_50_ against DENV-2: 67.1 μM	[126]
Doxycycline			IC_50_ against DENV-2: 55.6 μM	
A5			IC_50_ against DENV-2: 1.2 ± 0.7 μM	[127]
Compound 6			EC_50_: D1-0.108 ± 0.08 μM D2- 0.068 ± 0.01 μM D3-0.496 ± 0.09 μM D4-0.334 ± 0.12 μM	[128]
P02	E protein	Inhibition of virus entry	N/D	[129]
gg-ww			IC_50_ against DENV-2: 77 and 91 μmol L^−1^, determined using plaque assay and RT-PCR respectively.	[130]
EF		Inhibition of virus binding and entry	IC_50_ against DENV-2: 96 μM	[108]
Geraniin			IC_50_ against DENV-2: 1.75 μM	[50,131]
DET2			IC_50_ against DENV-2: >500 μM	[132]
DET4			IC_50_ against DENV-2: 35 μM	
Peptide 1			N/D	[133]
MLH40	prM/M protein	Inhibition of interactions between DENV M and E proteins	IC_50_: D1-30.35 ± 1.25 μM D2-31.41 ± 1.09 μM D3-27.95 ± 1.41 μM D4-24.45 ± 1.20 μM	[115]
pr		Fusion inhibitor	N/D	[116]
Pep14-23	C protein	Inhibition of interactions between the DENV C protein and host intracellular lipid droplets	EC_50_ against DENV-2: 0.016 μM	[119]
VGTI-A3			IC_90_ against DENV-2: 112 nM	[122]
VGTI-A3-03			IC_90_ against DENV-2: 25 nM	

N/D = Not determined. The IC_50_ value referred to the half-maximal inhibitory concentration required to inhibit the virus. The IC_90_ value referred to the concentration of the compound required to inhibit viral replication by 90%. The EC_50_ value referred to the concentration of the compound that gave the half-maximal response.

#### 5.2.3. Targeting DENV Non-Structural Proteins

Inhibition of DENV non-structural proteins may also lead to the inhibition of various post-infection stages such as viral replication, assembly, maturation, and polyprotein cleavage due to their vital roles as components of the DENV replication machinery. The NS3 and NS5 proteins are the most extensively studied non-structural proteins, and they are regarded as the most important targets for development of DENV antivirals due to their enzymatic activities during DENV replication [4]. 

##### Targeting NS1 Protein

The NS1 protein is 46–55 kDa in size, depending on its glycosylation state. It can be found in various cellular locations and has several oligomeric forms, including ER-resident form, secreted form, and membrane-anchored form [134]. Intracellular NS1 is involved in early DENV viral RNA replication. NS1 is usually transported to the cellular surface for membrane association or secretion as a soluble hexamer [135]. Studies have revealed that NS1 is greatly immunogenic which makes it a good target for vaccine development [134,136,137].

Limited inhibitors that target the NS1 protein have been developed and investigated for their anti-DENV activities (Table 4). Songprakhon et al. (2020) identified four peptides (peptides 3, 4, 10, and 11) which effectively inhibited DENV infections by binding to the DENV NS1 protein. It was revealed that these peptides could bind to the DENV-2 NS1 protein spontaneously due to their highly negative binding free energy values. It was found that all four peptides (10 μM) significantly reduced the amount of DENV-2 virions (42–57%) at 4 h post-infection. When the concentration was increased to 20 μM, a greater reduction in the production of DENV-2 virions (69–79%) was observed at 4 h post-infection. Peptide 3 was the most effective in lowering the production of DENV-2 virions, but it also caused the highest amount of cell death when administered at 20 μM. Apart from DENV-2, peptides 3 and 4 also effectively reduced the production of DENV-1 (62%) and DENV-4 (64%) virions, respectively, during post-infection. However, none of the peptides could inhibit DENV-3 production [138]. In another study by Lee et al. (2017), it was found that honeysuckle (*Lonicera japonica* Thunb.) extracts induced the expression of miRNA let-7a both in vitro and in vivo. Additionally, miRNA let-7a was found to target nucleotides 3313–3333 of the conserved NS1 sequence of DENV-1, DENV-2, and DENV-4. This resulted in the suppression of NS1 protein expression and viral replication. ICR-suckling mice treated with honeysuckle aqueous extract pre- and post-DENV-2 infection exhibited reduced levels of NS1 RNA and protein expression as well as reduced viral loads, prolonged survival time, and alleviated disease symptoms [139].

##### Targeting NS3 Protein

The 69 kDa NS3 protein is the second largest DENV protein, and it has various enzymatic activities such as 5′-RNA triphosphatase, nucleoside triphosphatase (NTPase), helicase, and serine protease. It consists of a protease domain at the N-terminal which cleaves the viral polyprotein into separate individual proteins and a RNA helicase domain at the C-terminal with vital roles in viral RNA replication and synthesis. The NS2B cofactor is required by the NS3 protease for proper protein folding and enzymatic activity [140,141]. The NS3 RNA helicase domain participates in viral RNA replication with other DENV non-structural proteins such as NS5. The helicase activity is also vital for the fusion of secondary structures at the UTRs before the start of DENV RNA synthesis. It also functions to unwind the dsRNA intermediate products generated during DENV RNA synthesis, before capping of the positive strand RNA [141].

Most compounds that were identified as DENV NS3 inhibitors were targeting the NS3 protease, probably due to the lack of binding pockets on the NS3 helicase for inhibition [142,143]. However, several inhibitors of DENV NS3 helicase have been developed and evaluated for their anti-DENV activities (Table 4). Ivermectin effectively inhibited DENV-2 NS3 helicase with an IC_50_ value of 500 ± 70 nM. Its mechanism of inhibition for DENV-2 NS3 helicase was found to be non-competitive with an inhibition constant of 354 ± 23 nM, which indicated that ivermectin was able to effectively bind to NS3 helicase when RNA was present [144]. A benzoxazole compound, ST-610, was also found to inhibit DENV NS3 helicase activity. ST-610 exhibited potent inhibitions of DENV-1 to 4 in vitro with EC_50_ values ranging from 0.203–0.272 μM. Time-of-addition studies revealed that this compound was most effective when added to DENV-2-infected cells within 4 h post-infection, indicating a post-entry mechanism of action. It was also found that the A263T mutation in the NS3 helicase domain conferred resistance to ST-610, which suggested that it targeted the DENV NS3 helicase. A molecular-beacon-based helicase assay confirmed that ST-610 inhibited the unwinding activity of the DENV NS3 helicase due to reductions in the levels of fluorescence quenching in a dose-dependent manner [145].

Various inhibitors that target the DENV NS2B-NS3 protease have been developed and investigated for their anti-DENV activities (Table 4). Rothan et al. (2012) revealed that protegrin-1 (40 μM) significantly reduced the DENV protease activity by 95.7% [146]. Apart from that, a disulphide cyclic peptide, retrocyclin-1 (100 μM), demonstrated 100% inhibition of the activity of DENV-2 protease with the lowest IC_50_ value of 14.1 ± 1.2 μM reported at 40 °C [147]. The repurposed drug, nelfinavir, exhibited good anti-DENV-2 activity with an EC_50_ value of 3.5 ± 0.4 μM and a selectivity index of 4.6. It was also demonstrated that nelfinavir was able to bind to the active site of the DENV-2 NS2B-NS3 protease by forming hydrophobic and polar interactions. Nelfinavir formed backbone hydrogen bonds with Met84 and Gly153, side chain hydrogen bond with Thr83, and π-π stacking interaction with Tyr161 [148].

##### Targeting NS4 Protein

The highly hydrophobic NS4A protein acts jointly with other viral and host proteins to promote rearrangements of membranes which are essential for viral replication [149,150,151]. Residues 1–49 of NS4A were found to act as a cofactor of the NS3 protein, residues 50–73, 76–89, and 101–127 were hydrophobic regions associated with membranes, while residues 123–130 constituted a small loop which exposed the cleavage site of NS4A-2k. The C-terminal 2k fragment functioned as a signal sequence vital for translocating the NS4B protein into the ER lumen [149]. On the other hand, the highly hydrophobic NS4B protein was able to inhibit STAT1 phosphorylation by blocking the signal transduction cascade induced by interferon-α/β [152]. NS4B protein was reported to modulate the function of the NS3 helicase domain, depending on its conformation. NS4A and NS4B were also shown to act cooperatively during viral replication [149,153,154]. 

Several inhibitors that target the NS4 protein have been developed and investigated for their anti-DENV activities (Table 4). A study by van Cleef et al. (2016) revealed the ability of the paracetamol metabolite, AM404, to block DENV replication by targeting the NS4B protein. Treatment of DENV-2-infected HeLa cells with AM404 resulted in 3- and 25-fold reductions of DENV-2 viral RNA at 48 and 72 h post-infection, respectively. Treatment with AM404 also resulted in 16- and 19-fold reductions of DENV-1 viral RNA at 48 and 72 h post-infection, respectively, whereas only two-fold reductions of DENV-4 viral RNA were observed at both time points. These results implied that AM404 inhibited a post-infection stage of the DENV replication cycle. Since AM404 strongly reduced the luciferase activity at 48 h post-transfection but not at 8 h post-transfection, this indicated that AM404 inhibited viral RNA replication and not viral translation. Mutations in the DENV NS4B protein were also found to render the virus insensitive to AM404, which suggests that AM404 might directly target the NS4B protein, though further studies were required to confirm these interactions [155]. Wang et al. (2015) demonstrated that a spiropyrazolopyridone compound, compound **1a**, selectively inhibited DENV-2 and DENV-3 with EC_50_ values ranging from 10–80 nM by targeting the NS4B protein. It was found that variations at amino acid 63 of the NS4B protein contributed to its selective inhibition of DENV-2 and DENV-3. The physicochemical properties of compound 1a were further improved to generate compound 14a which exhibited better aqueous solubility, though its anti-DENV-2 and DENV-3 activities were slightly reduced. The coelution of ^3^H-labeled compound **14a** with the wild-type DENV-2 NS4B protein confirmed that this compound could directly bind to the NS4B protein. Additionally, compound 14a also showed good in vivo antiviral efficacy since treatment of DENV-2-infected AG129 mice with compound 14a resulted in reduced viremia [156]. Another compound, NITD-618, specifically inhibited DENV-1 to 4 with EC_50_ values ranging from 1–4 μM and mode-of-action studies confirmed that this compound inhibited DENV RNA synthesis. It was found that P104L and A119T mutations in the DENV NS4B protein conferred resistance to NITD-618 which suggests that this compound targeted the NS4B protein, though further studies are required to confirm these interactions [51].

##### Targeting NS5 Protein

The 100 kDa NS5 protein is the largest DENV non-structural protein which shares 75% homologous sequences among all four DENV serotypes [157,158]. It forms part of the DENV viral replication complex and has various biological and enzymatic functions. NS5 consists of a methyltransferase (MTase) domain at the N-terminal, which is involved in the synthesis and methylation of the 5′ RNA cap, and a RNA-dependent RNA polymerase (RdRp) domain at the C-terminal which is involved in viral RNA synthesis. It plays an essential role in the DENV replication cycle, which makes it a good target for the development of anti-DENV therapeutics. NS5 interacts with the NS3 protein, host cellular proteins, and stem loop A (5′ end of DENV RNA genome) to promote viral RNA replication and synthesis [159,160]. A study by Yang et al. (2022) showed that I-OMe tyrphostin AG538 (I-OMe-AG238) and suramin hexasodium (SHS) were able to inhibit NS5-NS3 binding via direct binding to NS5. Both compounds disrupted NS5-NS3 binding with low IC_50_ values ranging from 3–6 μM. Nuclear localization of NS5 is a vital step in DENV infections, and it was mediated with the nuclear transport factor Importin (IMP) α and the IMPβ1 subunit; inhibiting this interaction might limit the production of infectious DENV particles. Both compounds were also found to inhibit NS5:IMPαΔIBB interaction which implied that they could bind directly to the NS5 protein. This was confirmed via thermostability analysis as the presence of these compounds altered the thermostability of the NS5 protein [161]. NS5 also carries out biological functions such as promoting hSTAT2 degradation to suppress type I interferon response [162].

Limited inhibitors that targeted the DENV NS5 MTase to block DENV infection have been identified (Table 4). An adenosine derivative, cordycepin, demonstrated significant inhibition of DENV-2 E protein during post-infection with an EC_50_ value of 26.94 μM. Since cordycepin was able to inhibit DENV-2 RNA replication, the authors hypothesized that it might be able to bind to DENV NS5 which is a vital enzyme in RNA synthesis. It was shown that cordycepin was able to bind to DENV NS5 MTase at a pocket on the SAM site. The conversion of SAM into SAH by NS5 MTase during the methylation process could be vital for DENV RNA replication. In addition, cordycepin was also able to bind to DENV NS5 RdRp. It was proposed that the binding of cordycepin to these two DENV NS5 domains strongly inhibited DENV RNA replication (Figure 4) [163].

Apart from DENV NS5 MTase, various inhibitors that target the DENV NS5 RdRp have also been identified (Table 4). Coulerie et al. (2013) demonstrated that extracts obtained from *Myrtopsis corymbosa* of the Rutaceae family strongly inhibited the DENV-2 NS5 RdRp. The crude ethyl acetate leaf (L1) and bark (B1) extracts exhibited significant inhibitions of the DENV-2 NS5 RdRp by 78% and 92%, respectively, when administered at 10 μg/mL. Additionally, when B1 extract was administered at 1 μg/mL, it inhibited DENV-2 NS5 RdRp by 87%. The dichloromethane and alkaloidal extracts of leaves (L2 and L3) and barks (B2 and B3) also exhibited good inhibitions against DENV-2 NS5 RdRp and were further purified to determine their constituents. Compounds **1**, **2**, and **3** which were isolated from B2 extract were identified as ramosin, myrsellinol, and myrsellin, respectively. Compounds **4**, **5**, and **6** which were isolated from L3 extract were identified as skimmianine, γ-fagarin, and haplopin, respectively. These isolated compounds demonstrated weak antiviral activities against DENV-2 NS5 RdRp when tested separately which suggested that their antiviral activities could be due to other minor compounds in the extracts or in synergy with other compounds [164]. A small molecule compound, RK-0404678, effectively inhibited DENV-1 to 4 NS5 RdRp with IC_50_ values ranging from 46.2–445 μM. RK-0404678 was found to bind to two distinct sites on the DENV-2 NS5 RdRp, which were conserved in the flavivirus NS5 proteins. In site 1, the benzoxathiole ring of RK-0404678 was surrounded by Ser763, Cys780, Asp808, Met809, and Tyr882 whereas the acetate group interacted with Arg773, Asn777, Trp833, and Met883. In site 2, the benzoxathiole ring of RK-0404678 interacted with Val507, Glu510, Gly511, Ser661, and Cys709. The binding of this compound induced a conformational change in the NS5 RdRp whereby Tyr607 on the α16 helix which faced the catalytic center in the absence of the compound was flipped in the opposite direction. This change caused a disorder, thereby affecting the structures of the α16 helix and its flanking loop, and inhibited DENV viral replication [165].

**Table 4 viruses-15-00705-t004:** Direct-acting antivirals of DENV targeting DENV non-structural proteins.

Drug	Target(s)	Mechanism(s) of Action	Inhibitory Activities (IC_50_/EC_50_/EC_90_ Values)	Reference
Peptide 3	NS1 protein	NS1 inhibition	N/D	[138]
Peptide 4				
Peptide 10				
Peptide 11				
Honeysuckle (*Lonicera japonica* Thunb.) extracts		Inhibition of NS1 protein expression and viral replication		[139]
Ivermectin	NS3 helicase and NS2B-NS3 protease	NS3 helicase and NS2B-NS3 protease inhibition	IC_50_ against DENV-2: 0.50 ± 0.07 μM EC_50_ against DENV-2: 0.70 μM	[144,166]
ST-610	NS3 helicase	NS3 helicase inhibition	EC_50_ against DENV-2: 0.272 μM EC_90_ against DENV-2: 3.59 μM	[145]
Suramin			IC_50_ against DENV-4: 0.80 μg/mL	[167]
Compound 25			IC_50_ against DENV-2: 78 ± 23 μM EC_50_ against DENV-2: 36 ± 6 μM	[168]
Compound 7			IC_50_ against DENV-2: 6 ± 5.4 μM	[169]
Protegrin-1	NS2B-NS3 protease	NS2B-NS3 protease inhibition	IC_50_ against DENV-2: 11.7 μM	[146]
Retrocyclin-1			IC_50_ against DENV-2: 21.4 μM at 37 °C and 14.1 μM at 40 °C	[147]
Nelfinavir	NS2B-NS3 protease	NS2B-NS3 protease inhibition	EC_50_ against DENV-2: 3.5 ± 0.4 μM	[148]
Carnosine			IC_50_ against DENV-2: 63.7 μM	[170]
Palmatine			N/D	[171]
Thiazolidinone-peptide hybrids				[172]
Compound 32				[173]
Compound 1			IC_50_: D1-36.4 μM D2-6.0 ± 2.6 μM D3-17.5 μM D4-32.8 μM	[174]
166347			IC_50_: D1-3 ± 1 μM D2-5 ± 2 μM D3-5 ± 2 μM D4-11 ± 3 μM	[175]
ARDP0006			EC_50_ against DENV-2: 4.2 ± 1.9 μM	[176]
ARDP0009			EC_50_ against DENV-2: 35 ± 8 μM	
Compound 7n			IC_50_ against DENV-2: 3.75 ± 0.06 μM	[177]
Diaryl(thio)ethers	NS2B-NS3 protease	NS2B-NS3 protease inhibition	IC_50_: D2-4.2–98 μM D3-0.99–31.8 μM	[178]
Compound C			IC_50_: D1-4.06 ± 0.21 μM D2-4.05 ± 0.18 μM D3-2.94 ± 0.18 μM D4-3.40 ± 0.11 μM	[179]
Compound D			IC_50_: D1-10.83 ± 0.37 μM D2-10.45 ± 0.40 μM D3-11.14 ± 0.38 μM D4-11.04 ± 0.37 μM	
Compound F (tolcapone)			IC_50_: D1-1.15 ± 0.1 μM D2-0.98 ± 0.06 μM D3-0.91 ± 0.06 μM D4-0.64 ± 0.03 μM	
SK-12			IC_50_ against DENV-1 to 4: 0.74–4.92 μM	[180]
Compound 104			IC_50_ against DENV-2: 0.176 μM	[181]
Ltc1	NS2B-NS3 protease	NS2B-NS3 protease inhibition	IC_50_ against DENV-2: 12.68 ± 3.2 μM at 37 °C and 6.58 ± 4.1 μM at 40 °C	[182]
BP13944			IC_50_ against DENV-2: 22.63 ± 0.74 μM EC_50_ against DENV-2: 0.23 ± 0.01 μM	[183]
Policresulen			IC_50_ against DENV-2: 0.48 μg/mL	[184]
BP2109			IC_50_ against DENV-2: 15.43 ± 2.12 μM EC_50_ against DENV-2: 0.17 ± 0.01 μM	[185]
MB21			IC_50_ against DENV-2: 5.95 μM	[186]
Compound 45a			IC_50_ against DENV-2: 0.26 ± 0.03 μM	[187]
Compound 14			N/D	[188]
AM404	NS4B	NS4B inhibition	EC_50_ against DENV-2: 3.6 μM	[155]
Compound 1a			EC_50_: D1->1 μM D2-0.012 ± 0.004 μM D3-0.032 ± 0.011 μM D4->1 μM	[156]
Compound 14a	NS4B	NS4B inhibition	EC_50_: D1->20 μM D2-0.042 ± 0.016 μM D3-0.076 ± 0.019 μM D4->20 μM	[156]
NITD-618			IC_50_: D1-1.5 μM D2-1.6 μM D3-1.6 μM D4-4.1 μM	[51]
AZD0530			N/D	[189]
Dasatinib				
JNJ-1A			EC_50_ against DENV-1, 2, and 4: ~1 μM	[190]
NITD-688			N/D	[191]
JNJ-A07		Inhibition of interactions between NS3 and NS4B proteins	EC_50_ against DENV-2: 0.035 μM	[192]
Compound B	NS4A	Inhibition of viral replication	IC_50_: D1-1.81 μM D2-1.32 μM D3-2.66 μM D4-4.12 μM	[193]
Cordycepin	NS5 MTase and NS5 RdRp	Inhibition of viral replication	EC_50_ against DENV-2: 26.94 μM	[163]
Azidothymidine-based triazoles	NS5 MTase	Inhibition of viral RNA capping	EC_50_ against DENV-2: 7.3-14 μM	[194]
Compound 10			N/D	[195]
BG-323				[196]
NSC 12155			EC_50_ against DENV-2: 7.0 μM	[197]
*Myrtopsis corymbose* extracts	NS5 RdRp	NS5 RdRp inhibition	N/D	[164]
RK-0404678			IC_50_: D1-46.2 ± 2.8 μM D2-201 ± 4.9 μM D3-287 ± 11 μM D4-445 ± 23 μM EC_50_: D1-29.5 ± 4.2 μM D2-6.0 ± 0.30 μM D3-29.4 ± 1.8 μM D4-31.9 ± 2.8 μM	[165]
Trigocherrins	NS5 RdRp	NS5 RdRp inhibition	IC_50_ against DENV-2: 3.1-16 μM	[198]
Trigocherriolides				
Chartaceones			IC_50_ against DENV-2: 1.8-4.2 μM	[199]
Avicularin			IC_50_ against DENV-2: 1.7 μM	[200]
Quercitrin			IC_50_ against DENV-2: 2.1 μM	
Betulinic acid			IC_50_ against DENV-2: 1.7 μM	
Spiraeoside			IC_50_ against DENV-2: 1.9 μM	
Rutin			IC_50_ against DENV-2: 2.1 μM	
Pyridobenzothiazolones			IC_50_ against DENV-2: 9.164-81.29 μM EC_50_ against DENV-2: 1.8-3.7 μM	[201]
(E)-tridec-2-en-4-ynedioic	NS5 RdRp	NS5 RdRp inhibition	IC_50_ against DENV-2: ~3 μM	[202]
Octadeca-9,11,13-triynoic acid				
Octadic-13-en-9,11-diynoic acid				
Octadic-13-en-11-ynoic acid				
7-deaza-2′-C-methyl-adenosine		Inhibitor of viral replication	EC_50_ against DENV-2: 15 μM	[203]
INX-08189			N/D	[204]
BCX4430			EC_50_ against DENV-2: 32.8 μM EC_90_ against DENV-2: 89.3 μM	[205]
Balapiravir			N/D	[206]
NITD008			IC_50_ against DENV-2: 0.31 μM	[207]
2′-C-methylcytidine			IC_50_ against DENV-2: 11.2 ± 0.3 μM	[208]

N/D = Not determined. The IC_50_ value referred to the half-maximal inhibitory concentration required to inhibit the virus. The EC_50_ value referred to the concentration of the compound that gave the half-maximal response. The EC_90_ value referred to the concentration of the compound that gave 90% maximal response.

**Table 5 viruses-15-00705-t005:** Chemical structures of selected direct-acting antivirals of DENV.

Drug	Structure	Class of Compound	Reference
Compound **6**	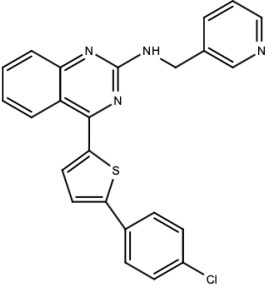	Thiophene pyrimidine	[128]
Compound **25** (PubChem CID: 45382104)	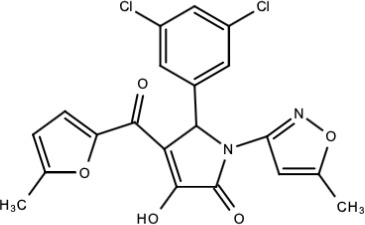	Pyrrolone	[168]
Compound **7**	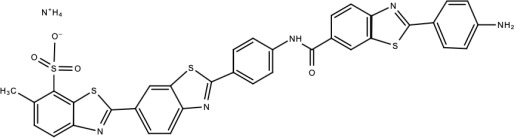	Benzothiazole	[169]
Compound **32**	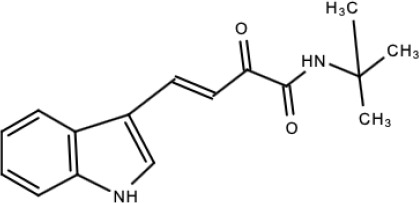	α-ketoamides	[173]
Compound **1**	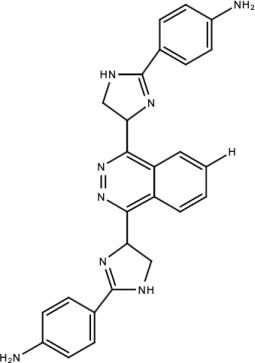	Phthalazine	[174]
Compound **7n**	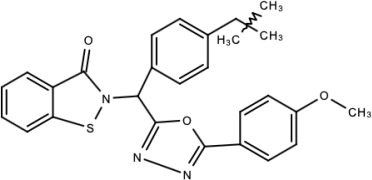	1,2-benzisothiazol-3(2H)-one—1,3,4-oxadiazole hybrid derivative	[177]
Compound C	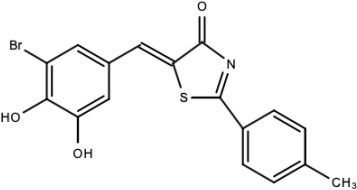	Catechols	[179]
Compound D	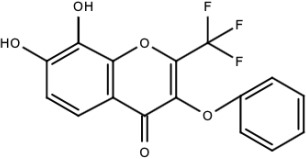
Compound F (tolcapone)	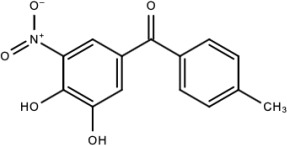	Catechols	[179]
Compound **104**	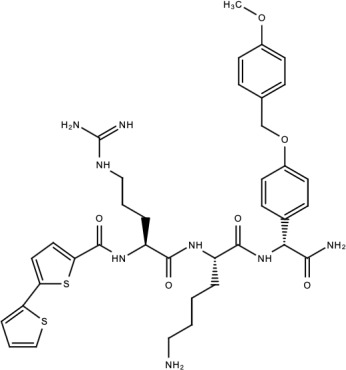	Bithiophene cap, 3-OCH_3_-benzyl ether	[181]
Compound **45a**	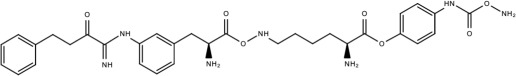	Phenylalanine-phenylglycine analogues	[187]
Compound **14**	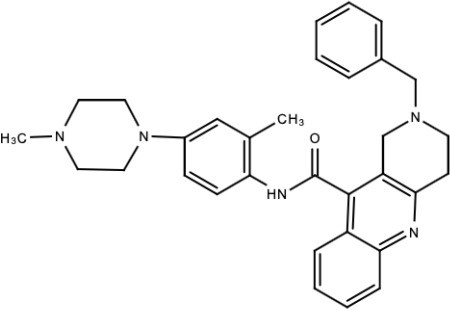	N/S	[188]
Compound **1a**	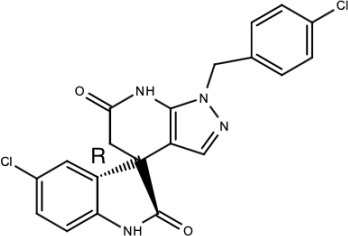	Spiropyrazolopyridone	[156]
Compound **14a**	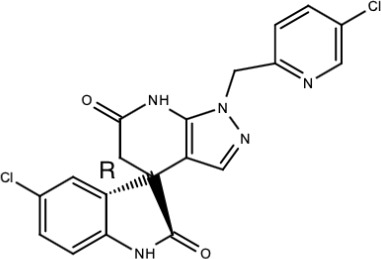
Compound B	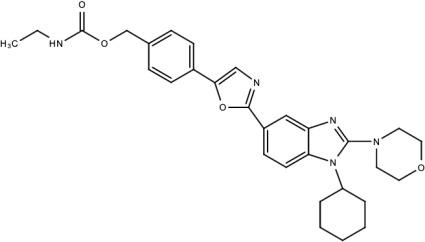	Benzimidazole	[193]
Compound **10**	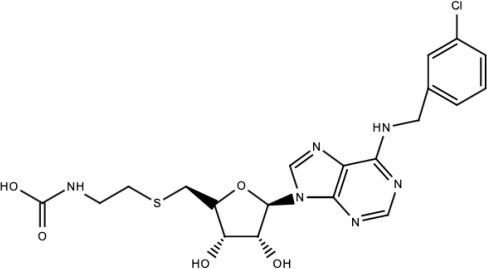	*S*-adenosyl-homocysteinederivatives	[195]

N/S = Not stated.

### 5.3. Targeting during Post-Infection Stages

Another approach which has been utilized to inhibit DENV infections is targeting the various stages during post-infection such as viral replication, viral maturation, and viral assembly. Table 6 summarizes DENV antivirals that target various post-infection stages.

#### 5.3.1. Inhibition of Viral RNA Synthesis and Viral Translation

RNA synthesis is a vital step for viral replication and a common target for antiviral drug development. Studies on DENV and other flavivirus replication complexes showed that viral replication occurred together with ER-derived cellular membranes. First, a negative strand intermediate is synthesized with the positive-sense genomic DNA strand as a template. This is followed by the synthesis of the positive strand DENV genomic RNA by a replicative intermediate complex with the resulting double-stranded RNA intermediate as a template. Viral replication proceeds on the replicative intermediate complex semi-conservatively and asymmetrically [209].

NS3 and NS5 proteins have been reported to play essential roles in viral RNA synthesis. The NS3 helicase domain participates in DENV RNA replication and synthesis together with NS5. NS3 helicase activity is vital for the fusion of secondary structures at the UTRs before the start of DENV RNA synthesis [141]. It also functions to unwind the dsRNA intermediate products formed during DENV RNA synthesis, before capping of the positive strand RNA [141]. On the other hand, the NS5 MTase domain is involved in 5′ RNA cap synthesis and methylation, and its RdRp domain is involved in viral RNA synthesis [159]. Therefore, inhibiting these DENV non-structural proteins might lead to the inhibition of viral replications. 

Rothan et al. (2012) showed that protegrin-1 significantly lowered the DENV-2 RNA copy number in a dose-dependent manner. Almost 100% reduction of DENV-2 RNA copy number was observed when MK2 cells were treated with 12.5 μM of protegrin-1. The protegrin-1 peptide also inhibited DENV-2 protease activity effectively with a *K_i_* value of 5.85 μM [146]. Subsequently, Rothan et al. (2014) reported that another peptide, the latarcin 1 peptide, was able to reduce the amount of DENV-2 RNA copies effectively with EC_50_ values of 8.3 ± 1.2 μM, 7.6 ± 2.7 μM, and 6.8 ± 2.5 μM after treatment with latarcin 1 for 24 h, 48 h, and 72 h, respectively. In addition, the latarcin 1 peptide also inhibited the activity of DENV-2 protease effectively with IC_50_ values of 12.68 ± 3.2 μM and 6.58 ± 4.1 μM at 37 °C and 40 °C, respectively [182]. Compound 1 (7-deaza-2′-C-acetylene-adenosine) was found to exert its anti-DENV activity through RNA chain termination based on three lines of evidence. First, compound 1 was able to suppress viral RNA synthesis but not viral translation. Next, the triphosphate form of compound 1 directly inhibited RNA elongation which was catalyzed by DENV NS5. Lastly, cocrystallization of the complex of DENV RdRp and the triphosphate form of compound 1 revealed that the triphosphate moiety was located at the active site of DENV RdRp [210]. Mycophenolic acid (MPA) was demonstrated to exert its anti-DENV activity by preventing viral RNA synthesis and accumulation of viral RNA. Treatment of Hep3B cells with MPA at 4 or 24 h post-DENV-2 infection resulted in a marked reduction of viral antigen expression by 99% and 94%, respectively. Treatment of Hep3B cells with MPA at 4 or 24 h post-DENV-2 infection also reduced the production of infectious DENV-2 particles by 6 and 3 log reduction, respectively. Since MPA was able to block the accumulation without promoting the degradation of viral RNA in DENV-2 infected cells or DENV-2 RNA transfected cells, it was hypothesized that MPA possibly inhibited viral RNA synthesis or the viral translation step. Since treatment with MPA did not inhibit the early phase of viral translation, the mechanism of action of MPA was determined to be an inhibition of viral RNA synthesis due to its ability to inhibit the accumulation of both positive- and negative-strand viral RNA [211].

DENV utilizes the host cell translation machinery to produce a polyprotein which undergoes proteolytic cleavage to generate DENV structural and non-structural proteins. The translation process occurs primarily via cap-dependent initiation whereby the DENV 5′ type 1 7-methylguanosine cap is recognized and bound by the eukaryotic translation initiation factor (eIF) 4E [212,213,214]. Next, eIF4E recruits the eIF4G scaffolding protein and eIF4A helicase to form a complex. The association of eIF3 with this complex mediates the binding of 43S ribosomal pre-initiation complex with 60S subunit to form the 80S ribosomal complex which is competent for elongation. The 3′-poly(A) tail functions to recruit the poly(A)-binding protein (PABP) which binds to eIF4G for mRNA circularization and recruitment of ribosomes [209].

A study by Wang et al. (2011) reported that a benzomorphan compound, NITD-451, inhibited DENV infection via suppression of the viral translation process. Its mechanism of action was determined to be translation inhibition based on five lines of evidence. First, the time-of-addition experiments revealed that upon addition of NITD-451 after 16 h post-transfection, no anti-DENV effects were observed which indicated that NITD-451 targeted an early stage of DENV infection. Next, NITD-451 effectively inhibited replication of DENV-1 replicon as much as a complete virus infection which indicated that it did not inhibit viral entry and assembly. ATPase, protease, MTase, and RdRp assays were performed with NITD-451 and no inhibition was observed. In addition, results from the replicon transiently expressing DENV-1 genes showed that NITD-451 inhibited the translation of input RNA inside the cells. Lastly, it was observed that NITD-451 could also suppress the translation of a reporter RNA in vitro [215]. Another study by Low et al. (2011) discovered that an ionophore, narasin, inhibited DENV infection by blocking viral protein synthesis. Narasin potently inhibited DENV-1 to 4 with IC_50_ values ranging from 0.05–0.65 μM. It was found that treatment of Huh-7 cells with narasin upon transfection of DENV-2 viral RNA completely inhibited DENV-2 replication. This indicated that narasin exerted its antiviral activity at a step following release of DENV viral RNA into the cytoplasm for viral replication. Treatment of DENV-2-infected cells with narasin from 12 to 48 h post-infection did not reduce the amount of positive- and negative-strand DENV-2 RNA, which indicated that narasin did not inhibit DENV RNA synthesis. However, the expressions of DENV E and NS5 proteins were significantly reduced when DENV-2 infected cells were treated with narasin at 12 h post-infection, thereby indicating that narasin exerted its antiviral activity by inhibiting viral protein synthesis [216]. Other compounds such as lactimidomycin, ST081006, bromocriptine, and peptide-conjugated phosphorodiamidate morpholino oligomers were also identified to be translation inhibitors of DENV [217,218,219,220].

Since both the 3′UTR and 5′-UTR are vital for viral replication, inhibition of these UTRs may inhibit DENV infection. Currently, there are no known antiviral peptides that could inhibit the viral replication process by targeting the 3′-UTR or 5′-UTR. Various miRNAs such as miR-133a, miR-744, miR-548 g-3p, and miR-484 have been reported to inhibit DENV infection by targeting the 3′-UTR or 5′-UTR [221,222,223]. It was demonstrated that miR-133a impaired DENV replication by targeting the 3′-UTR in a study by Castillo et al. (2016). Bioinformatic analysis revealed that miR-133a was able to bind directly to a sequence in the 3′SL loop of the 3′-UTR. The 3′SL loop contained 3′CS and 3′UAR elements which were vital for genome circularization and viral viability. Additionally, overexpression of miR-133a resulted in a significant reduction in the expression of PTB cellular protein at 12 h post-infection, but an increase in the expression of PTB was observed at 24 and 48 h post-infection. The PTB cellular protein was identified to be a vital component during DENV replication, due to its ability to interact with DENV UTRs and its role in pre-mRNA processing, regulation of polyadenylation, and viral translation. The authors proposed that the PTB cellular protein played a vital role during the first few hours of DENV replication. Low levels of PTB during the first few hours of DENV replication potentially altered DENV RNA circularization, which was essential for viral RNA synthesis and translation [222]. Similar to miR-133a, it was demonstrated that miR-484 and miR-744 blocked DENV replication by targeting a sequence in the 3′SL loop of the 3′-UTR. The overexpression of these miRNAs resulted in suppression of DENV-2 NS1 production, which suggested that they may affect DENV replication. However, DENV infection also simultaneously reduced the expression of both miRNAs. The authors proposed that the interactions of these miRNAs with DENV 3′-UTR possibly blocked the recruitment of PTB, thus affecting DENV RNA circularization, which was essential for viral RNA synthesis and translation [223].

Host RNA-binding proteins such as AePur and AeStaufen found in the cells of *Aedes aegypti* mosquitoes were revealed to act as restriction factors which prevented DENV replication. AePur exerted its antiviral function by interacting with the DENV-2 3′-UTR whereas AeStaufen interacted with both the DENV-2 3′-UTR and subgenomic flaviviral RNA (sfRNA). Both human and mosquito Pur homologs demonstrated a shared affinity for DENV-2 RNA, but its antiviral function is only specific to the mosquito protein. AeStaufen mediated the reduction of RNA and sfRNA copies in the mosquito salivary glands and the reduction of sfRNA copies caused a significant drop in the amount of sfRNA which enhanced DENV transmission in the saliva [224]. Another study by Liao et al. (2018) demonstrated that a host RNA-binding protein, quaking protein (QKI), inhibited DENV-4 infection in Huh7 cells by interacting with the DENV-4 3′-UTR to suppress translation of viral genomes. However, QKI could only bind to DENV-4 3′-UTR and not the UTRs of the other three DENV serotypes [225]. Since these host RNA-binding proteins could exert antiviral functions by binding to DENV 3′-UTR, peptides designed to mimic these host RNA-binding proteins might also be able to inhibit DENV 3′-UTR.

#### 5.3.2. Inhibition of Virus Assembly, Maturation, and Release

Nucleocapsid formation or encapsulation of the viral genome is a form of electrostatic interaction which facilitates binding of the positive-charged C protein with the negative-charged viral RNA [226]. The interactions between 3′-UTR with residues R94, K95, and K99 of the NS2 protein mediated the binding of the viral positive-sense RNA to the NS2 protein [227]. The NS2 protein also functioned to recruit the translated polyproteins and NS2B-NS3 proteases to the site of assembly [227,228,229]. An anchor signal peptide spanning the ER membrane connected the C and prM proteins. Next, cleavage of the C-anchor junction by the NS2B-NS3 protease resulted in a free prM and mature C protein which formed the nucleocapsid [226,230]. The nucleocapsid, ER membrane, E and prM proteins travel together into the ER lumen and secretory pathway as an immature virion. The pr segment of the prM protein caps the fusion loop of the E protein to prevent premature virus–host cell membrane fusion [230,231].

The pH-triggered conformational changes of the E protein of the immature virion occurs after going through the secretory pathway during virus release. The E protein trimers form part of the immature virions whereas the E homodimers form part of the mature virus, and this is observed during the membrane fusion process [232]. The virions travel through the Golgi apparatus into the TGN where the cleavage of prM protein by host furin proteases takes place, which transitions the spiky immature virion into a smooth mature virion [112]. The pr segment continues to associate with the virion until its release from the host cell to prevent premature E protein-exosomal membrane binding [233]. In addition, pr is also needed for interactions with the vacuolar-ATPases which acidifies organelles and components of the secretory pathway for the processing and degradation of proteins during DENV infection, constituting a mechanism which favors the transport of viruses, providing stability in the host cell secretory pathway, and providing an optimal pH environment for efficient secretion of viruses [234]. The glycosylated states of residues 67, 153, and 154 of the DENV E protein are vital as loss of these glycans might cause a reduced release of virus from host cells. Mature virions released from the TGN enter the host cytoplasm, followed by exocytosis [233,235,236]. The cleaved off pr segment and mature virions are released into the extracellular fluid upon secretion of particles [233]. The mature DENV virion consists of a C protein encapsulating the positive-sense single-stranded RNA, a lipid bilayer containing transmembrane viral proteins, and a glycoprotein shell of E and M proteins [231].

A study by Martínez-Gutierrez et al. (2011) revealed that lovastatin inhibited DENV infection by targeting the viral assembly step. Time course studies revealed that lovastatin inhibited the final stages of the DENV replication cycle which was consistent with the build-up of viral RNA and proteins in lovastatin-treated DENV-2-infected cells during the post-infection stage. It was postulated that this occurred due to failure of viral RNA and proteins to undertake the normal assembly pathway [237]. In another study, it was reported that the c-Src protein kinase inhibitor, dasatinib, was able to block DENV virion assembly to inhibit DENV-2 infection. Treatment of DENV-2-infected cells with dasatinib did not affect the synthesis of DENV E proteins which indicated that it did not inhibit viral RNA synthesis and gene expression. Transmission electron microscopy (TEM) analysis revealed that there were no DENV virions present within the ER lumen of dasatinib-treated DENV-2-infected cells. Accumulation of nucleocapsid particles in virus-induced ER membranes was also observed within dasatinib-treated DENV-2-infected cells. Since the c-Src protein kinase mediated budding of the nucleocapsid into the ER lumen to form DENV viral particles, treatment with dasatinib prevented the viral assembly step [238]. Hirsutine, an indole alkaloid derived from *Uncaria rhynchophylla* was found to target the viral assembly or release step to block DENV infection. Hirsutine potently inhibited DENV-1 to 4 by 80–90% at a concentration of 10 μM. Time-of-addition and time-of-elimination studies demonstrated that hirsutine inhibited viral assembly, budding, or release but not viral RNA synthesis and translation [239]. Castanospermine also blocked DENV infection by targeting the viral release step. Several studies suggested that castanospermine might inhibit DENV infection by causing improper folding of DENV-1 E protein or by inhibiting interactions with the molecular chaperonin calnexin. Therefore, the authors proposed that this compound could inhibit virus morphogenesis and secretion of infectious DENV particles. Treatment of DENV-infected BHK-21 cells with castanospermine at 24 h and 48 h post-DENV infection markedly reduced the amounts of DENV viral RNA and infectious viral particles. At 48 h post-infection, the levels of DENV viral RNA and infectious viral particles were reduced by approximately 20-fold and 3000-fold, respectively. This increased the viral RNA-to-PFU ratio by approximately 150-fold, thereby suggesting that castanospermine reduced the infectivity of secreted DENV particles [240]. Brefeldin A (BFA), a fungal secondary metabolite from *Penicillium* sp., was proposed to block DENV infection by targeting the viral assembly, maturation, and release steps. BFA exhibited complete inhibition of DENV-2 when added at 24 h and 48 h post-infection with a concentration of 250 nM and good inhibitions of DENV-1 to 4 with EC_50_ values ranging from 54–65 nM. Time-of-addition studies revealed that addition of BFA at 18 h post-infection resulted in complete inhibition of DENV replication, which indicated that BFA inhibited viral assembly and release. Therefore, the authors proposed that BFA inhibited DENV maturation by blocking the trafficking of E proteins from the ER to the Golgi apparatus, leading to inhibition of virus formation and release [241].

**Table 6 viruses-15-00705-t006:** DENV antivirals targeting the post-infection stages.

Drug	Mechanism(s) of Action	Reference
Protegrin-1	Inhibition of viral RNA synthesis	[146]
Ltc 1		[182]
7-deaza-2′-C-acetylene-adenosine		[210]
Mycophenolic acid		[211]
NITD-451	Inhibition of viral translation	[215]
Narasin		[216]
Lactimidomycin		[217]
ST081006		[218]
Bromocriptine		[219]
Peptide-conjugated phosphorodiamidate morpholino oligomers		[220]
Lovastatin	Inhibition of viral assembly	[237]
Dasatinib		[238]
Hirsutine		[239]
Castanospermine	Inhibition of virus release	[240]
Brefeldin A	Inhibition of virus maturation and release	[241]

## 6. Combination Therapy

Treatment of diseases with combinations of two or more drugs is known as combination therapy. Combinations of drugs could result in several outcomes including synergistic and antagonistic effects as well as increased drug toxicity [242]. Although combinations of drugs could potentially cause adverse effects due to drug–drug interactions, it might also confer several advantages with careful use. Combination therapy could target multiple pathways to promote drug synergy, and these beneficial effects may outweigh its adverse effects. Synergistic drug combinations could increase treatment efficacy and reduce individual drug doses, thus resulting in increased tolerability in patients and reduced drug toxicities. Combination therapy could also delay the onset of drug resistance which is inevitable in certain diseases [243].

Combinations of several antiviral drugs have been tested against DENV both in vitro and in vivo. CM-10-18, a glucosidase inhibitor, only exhibited modest antiviral effects when evaluated against DENV-2 in vivo in AG129 mice. The combination of CM-10-18 and ribavirin, a broad-spectrum antiviral nucleoside analogue, was evaluated to determine whether this combination could synergistically inhibit DENV infections in vivo. Treatment of DENV-2-infected AG129 mice with 40 mg/kg of ribavirin by itself did not reduce viremia while treatment with 75 mg/kg of CM-10-18 by itself only modestly reduced viremia by 1.9-fold. However, the combination of both compounds caused a more significant reduction in viremia by 4.7-fold [244]. Franco et al. (2021) showed that UV-4B exerted potent inhibitory effects against DENV-2 in HUH-7, SK-N-MC, and HFF-1 cells with EC_50_ values of 23.75, 49.44, and 37.38 μM, respectively. On the other hand, interferon-alpha (IFN) also exerted potent inhibitory effects against DENV-2 in HUH-7, SK-N-MC, and HFF-1 cells with EC_50_ values of 102.7, 86.59, and 163.1 IU/mL, respectively. These two most potent anti-DENV agents were then subjected to combination therapy to determine whether this combination could synergistically inhibit DENV infections in vitro. The combination of UV-4B (25 μM) and IFN (100 IU/mL) reduced the viral titers by 3.5 log10 PFU/mL, whereas treatment of DENV-2-infected HUH-7 cells with each of the antiviral individually reduced the viral titers by 1.7 and 2.5 log10 PFU/mL, respectively. Similar observations were observed when DENV-2-infected SK-N-MC cells were treated with the combination of UV-4B (25 μM) and IFN (100 IU/mL). The viral titers were reduced by an additional 0.8 log10 PFU/mL when subjected to combination therapy in contrast to treatment with the antivirals individually. In DENV-2-infected HFF-1 cells, the combination of UV-4B (25 μM) and IFN (10 IU/mL) resulted in an additional ten-fold reduction in viral titers (2.3 log10 PFU/mL) when compared to treatment with the antivirals individually (1.3 log10 PFU/mL) [245].

Since there is a lack of clinically-approved DENV antivirals, combination therapy might be a good approach for the future development of DENV antivirals. Although studies have found synergistic drug combinations to be effective against DENV infections in vitro and in vivo, these combinations have not yet been clinically tested. It might also be advantageous to test combinations of antiviral drugs with different mechanisms of action to develop synergistic drug combinations for the treatment of dengue at any stage of the infection.

## 7. Conclusions

Due to the lack of efficacy of the Dengvaxia vaccine and the absence of clinically approved antivirals against DENV, the development of novel DENV antivirals is highly warranted. Various antiviral agents have been investigated for their anti-DENV activities and each antiviral employed a different mode of DENV inhibition. These antivirals might function by inhibiting the host cell receptors or attachment factors or by directly inactivating the virus by targeting the viral structural or non-structural proteins. Additionally, antivirals might also target different stages during post-infection such as viral replication by reducing the production of RNA copies, viral translation by reducing the expression of DENV viral proteins, or viral assembly, maturation, and release. More in-depth understanding of the molecular mechanisms of inhibition of the DENV life cycle by various compounds are vital to exploit their potential as antiviral candidates. The combinations of antivirals with different mechanisms of action could be evaluated to develop synergistic drug combinations for the treatment of dengue at any stage of the infection.

## Figures and Tables

**Figure 1 viruses-15-00705-f001:**
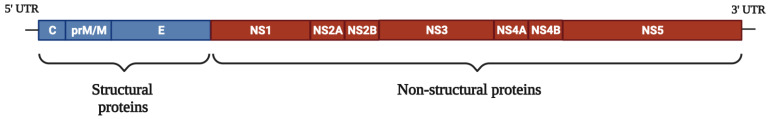
DENV RNA genome organization. The DENV positive-sense RNA genome consists of 11 kilobases (kb) with a single open reading frame (ORF) characterized by the presence of two untranslated regions (UTRs) at both ends of the ORF. The 5′-UTR consists of 95–135 nucleotides with a type I cap-like mRNA whereas the 3′-UTR consists of 114–650 nucleotides and lacks a poly(A) tail, ending in a conserved stem-loop secondary structure [23,24]. The figure was created using Biorender.com (ON, Canada) (accessed on 26 February 2023).

**Figure 2 viruses-15-00705-f002:**
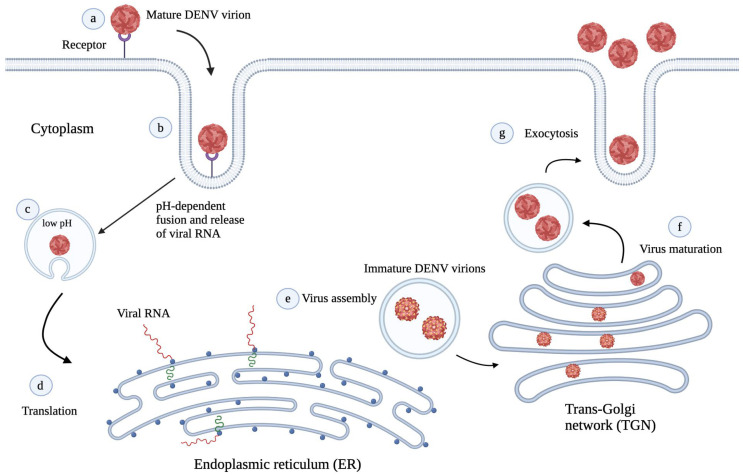
DENV life cycle. (**a**) DENV viral particles attach to the host cells via interactions between DENV surface proteins and their respective host receptors. (**b**) DENV enters host cells via receptor-mediated or clathrin-dependent endocytosis. (**c**) pH-dependent membrane fusion occurs in the endosome and this results in the formation of pores which allows the release of the DENV viral genome into the host cytoplasm. (**d**) The released viral RNA is translated into a polyprotein in the host cytoplasm. (**e**) Viral assembly takes place at the ER to form immature DENV virions. (**f**) These immature DENV particles travel through the secretory pathway and TGN to form mature DENV virions. (**g**) Following successful virus maturation, the mature DENV particles are then exocytosed from the host cells. The figure was created using Biorender.com (accessed on 7 March 2023).

**Figure 3 viruses-15-00705-f003:**
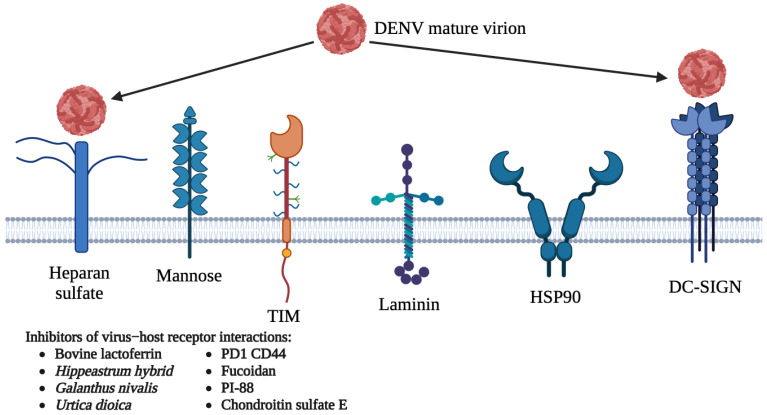
DENV host receptors as potential antiviral targets. Several molecules identified as possible host cellular receptors or attachment factors for DENV include heparan sulfate, mannose, DC-SIGN, laminin, HSP90/HSP70, and TIM and TAM proteins. The figure was created using Biorender.com (accessed on 26 February 2023).

**Figure 4 viruses-15-00705-f004:**
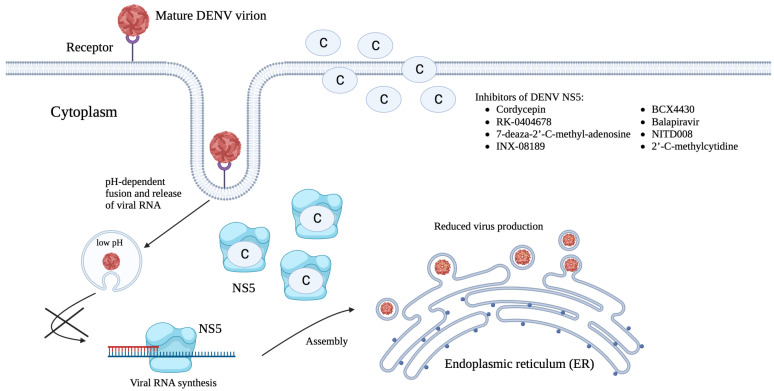
General molecular mechanism of compounds (labelled C) targeting DENV NS5. The binding of these compounds to DENV NS5 MTase and RdRp could possibly block the viral RNA synthesis step, resulting in inhibition of viral replication and reduced DENV production. The figure was created using Biorender.com (accessed on 7 March 2023).

**Table 1 viruses-15-00705-t001:** DENV antivirals that are currently undergoing clinical trials.

Drug	Clinical Trial Phase	Status	ClinicalTrials.gov Identifier *
JNJ-64281802	II	Recruiting	NCT05048875
Melatonin		Not yet recruiting	NCT05034809
AT-752	I	Recruiting	NCT05366439
Zanamivir		Not yet recruiting	NCT04597437

***** https://clinicaltrials.gov (accessed on 26 February 2023).

## Data Availability

Not applicable.

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
