# Peer review of "Molecular Mechanisms of Antiviral Agents against Dengue Virus"

_viruses, 2023, doi:10.3390/v15030705_

Round 1
Reviewer 1 Report
This is a well written review of the preclinical drug discovery strategies and targets for identifying antiviral agents for Dengue virus infection, with a comprehensive list of citations (246) representing the state of the art over the past 20+ years. This will be a useful review for investigators entering the field.
Author Response
There are no reviewer comments to be addressed.
Reviewer 2 Report
The authors make a well-researched analysis of the targets of the Dengue virus, but there are serious comments.
A significant number of reviews have been published in the last 10 years on the Dengue virus.
García, L.L., Padilla, L. & Castaño, J.C. Inhibitors compounds of the flavivirus replication process. Virol J 14, 95 (2017). https://doi.org/10.1186/s12985-017-0761-1
Guzman, M., Gubler, D., Izquierdo, A. et al. Dengue infection. Nat Rev Dis Primers 2, 16055 (2016). https://doi.org/10.1038/nrdp.2016.55
Harapan, H.; Michie, A.; Sasmono, R.T.; Imrie, A. Dengue: A Minireview. Viruses 2020, 12, 829. https://doi.org/10.3390/v12080829
J Pharm Pharm Sci . 2019;22(1):440-456. doi: 10.18433/jpps30216.
J. Med. Chem. 2016, 59, 12, 5622–5649
In order for the submitted review to be accepted for publication in a highly ranked journal, it is necessary to note what exactly makes this review different, what the emphasis is on and what exactly is new systematized.
There are a number of significant comments to the review presented.
For the most important point is the lack of structures for the compounds in question. Thus, more often than not, the authors cite in the text of the article such phrases - Many other E protein inhibitors such as geraniin, DV2419-447, DET2, DET4, P1, DN57opt, 1OAN1, rolitetracycline, doxycycline, A5, compound 6, P02, BP34610, compound 2, and gg-ww were also found to inhibit DENV infections [107-119].
What is compound 2, 6, etc.? In the original papers, the authors always depict the structures of compounds and assign them codes (or numbers, in which sequence they are mentioned). When reading this manuscript it is impossible to understand which chemical compounds we are talking about, you have to refer to the original works. As a result, there are phrases in the review text - compounds 2 (or another number) that refer to completely different substances, which is unacceptable.
Which peptides they are referring to? It might be useful to make a general table showing which peptides were considered to be inhibitors with references to the original papers.
The authors actively cite papers in which only molecular modelling has been carried out, without biological experiments. These are highly controversial papers, from the reviewer's point of view, without proper testing, the calculations themselves do not contribute meaningfully to the search for effective inhibitors and to actively cite such papers is only to promote 'information rubbish'.
In some cases, the authors write the activity EC50, and in some cases IC50. Either clarify what is meant or make it similar.
It is unacceptable to collect so many references in one sentence, not noting exactly what they refer to. For example, on page 17: Many other compounds such as carnosine, palmitine, thiazolidinone-peptide hybrids, compound 32, compound 1, 166347, ARDP0006, ARDP0009, compound 7n, diaryl(thio)ethers, compound C, compound D, compound F, compounds 1-6, compound 8, Ltc1, BP13944, policresulen, BP2109, MB21, compound 45a, compound 14, SK-12, compound 104, and ivermectin were also found to inhibit DENV infections by targeting the NS3 protease [139, 152-171]. How can the reader find the correct reference?
It is not very clear why Figure 3 is given specifically for the effects of cordycepin? If an analysis was carried out and the stages at which key potential antiviral agents acted were noted, it would be informative. In addition, this compound can play a role as a nucleoside inhibitor of replication, while the figure shows the post-endocytosis stage.
References must be formed according to the requirements of the journal.
Only after serious revision of the submitted manuscript can it be considered as a review in a highly prestigious journal.
Reviewer 3 Report
The significant increase in Dengue virus (DENV) infections over the last couple of decades together with questions about the efficacy and safety of the two commercially available DENV vaccines and the lack of approved DENV specific therapeutics, underscore the importance of identifying antiviral drug candidates that could be developed into safe and effective therapeutics. Hence, this review by Lee and colleagues, discussing the mechanisms by which compounds with reported anti-DENV activity inhibit DENV infection, addresses an important ant timely issue related to the ongoing threat posed by DENV o human health.
The authors have provided a comprehensive review of compounds and drugs that have been documented to target different steps of the DENV life cycle, with a focus on direct acting antivirals. This review represents a valuable contribution to the field and will be of interest to virologists working on both basic and clinical aspects of DENV infection.
The overall structure of the review is appropriate with exception of the discussion of strategies targeting non-structural proteins ahead of the section discussing targeting of viral activities taken place at a post cell entry stage of the infection.
The incorporation of a figure with a schematic of the DENV genome organization will be helpful. Likewise, the review will benefit from incorporating a table summarizing the most relevant antiviral drug candidates, their viral target and proposed mechanism of action.
The review should also incorporate a specific section about the preclinical status of the most advance antiviral drugs and their potential pathway to licensure. A discussion about combination therapy using existing DENV antiviral drug candidates would also add valuable information to the review.
The authors have used different font types and line numeration, likely the result of cutting and past. These editorial issues should be addressed.
Reviewer 4 Report
The manuscript “Molecular mechanisms of antiviral agents against dengue virus” by Lee, Wu and Poh is a well written, timely and comprehensive review about the current state of antiviral drug development to block DENV entry into host cells or virus replication. The development of antivirals targeting host receptors and DENV structural and non-structural proteins are summarized with a focus on replication, maturation and assembly of DENV.
Chapters 1-3 give a very good introduction regarding DENV structure, genome organization and its life cycle. Chapter 4 is subdivided into host-directed antivirals, direct-acting antivirals (targeting the viral structural or non-structural proteins) and targeting during post-infection stages. So far, no clinically approved antivirals against DENV are available.
Minor point
It would be helpful for the reader to include an additional Figure or a Table in the manuscript summarizing the most promising antiviral compounds, their targets, the respective EC50 values of the compounds and a reference for the compound.
Round 2
Reviewer 2 Report
The authors have made significant changes to the text of the review, but there are still comments to the review.
Figure 2 (Figure 1 in the previous version) depicts the life cycle of the virus. However, there are some errors in it. The stages d) and e) show translation and replication and the virus is depicted as a "balloon", when there is no virus particle as such. This scheme is depicted many times in reviews such as these:
https://pubs.acs.org/doi/10.1021/acsomega.2c00625
https://www.cell.com/fulltext/S0092-8674%2815%2900842-9
The authors prepared tables summarising the results, it became much more understandable. However, in terms of medicinal chemistry, it is still not clear what is compound 6 (Table 3), 25, 7, 32, 1 (Table 4), etc. Perhaps an additional table with structures of substances (or at least what class they belong to) should have been made.
Reviewer 3 Report
The authors have adequately addressed the concerns I raised during the review of the originally submitted paper. I do not have further comments regarding the scientific content of this revised version of the review by Lee and colleagues.
Author Response
There are no comments to be addressed from reviewer 3.